# Formic Acid Dehydrogenation over a Monometallic Pd and Bimetallic Pd:Co Catalyst Supported on Activated Carbon

María Ribota Peláez, E. Ruiz-López, M. I. Domínguez [ID], S. Ivanova [ID] and M. A. Centeno *[ID]

Instituto de Ciencia de Materiales de Sevilla (ICMS), Centro Mixto CSIC, Universidad de Sevilla, Avda. Américo Vespucio, 49, 41092 Sevilla, Spain; maria.ribota@icmse.csic.es (M.R.P.); eruizl@us.es (E.R.-L.); mdominguez1@us.es (M.I.D.); sivanova@us.es (S.I.)
* Correspondence: centeno@icmse.csic.es

**Abstract:** In this study, palladium is proposed as an active site for formic acid dehydrogenation reaction. Pd activity was modulated with Co metal with the final aim of finding a synergistic effect that makes possible efficient hydrogen production for a low noble metal content. For the monometallic catalysts, the metal loadings were optimized, and the increase in the reaction temperature and presence of additives were carefully considered. The present study aimed, to a great extent, to enlighten the possible routes for decreasing noble metal loading in view of the better sustainability of hydrogen production from liquid organic carrier molecules, such as formic acid.

**Keywords:** heterogeneous catalysis; hydrogen; bimetallic catalyst; formic acid dehydrogenation; Pd

## 1. Introduction

The increasing restrictions on greenhouse gas emissions, particularly $CO_2$, necessitate a change of the current energy model. The only possible green energy vector is hydrogen, and it presents multiple advantages. Contrary to fossil fuels, on which the current energy model is based, hydrogen is considered as a green vector, since its combustion produces energy without generating carbon-containing molecules and because of its possible generation from renewable sources [1]. Despite these key advantages, the high reactivity and low compression capacity of hydrogen make its handling and shipping difficult. In order to overcome these drawbacks, Liquid Organics Hydrogen Carriers (LOHCs) [2] arise as molecules capable of transporting hydrogen structurally, thus removing storage and transport problems associated with molecular hydrogen manipulation. Formic acid (FA) is one of the LOHCs that has attracted significant interest [3], being able to store 4.4 wt% of $H_2$ [4] and release it at relatively low temperatures via formic acid dehydrogenation reaction (FAD). The carbon dioxide discharged in this reaction can be re-hydrogenated to produce a new charge of formic acid, thus completing the cycle of loading ($CO_2$ hydrogenation to FA) and unloading (FAD) of hydrogen with zero net $CO_2$ emissions [5,6]. In addition, we can highlight the possibility of producing FA from renewable sources, e.g., biomass and glucose in particular [4,7,8].

Formic acid releases hydrogen through a dehydrogenation reaction (1), which competes in the gas phase with the dehydration one (2) [9], necessitating the search for a selective FAD catalyst.

$$HCOOH \rightarrow H_2 + CO_2 \tag{1}$$

$$HCOOH \rightarrow CO + H_2O \tag{2}$$

Both homogeneous catalysts such as Ru or Ir complexes [10–12] and heterogeneous ones based on noble metals such as Pd, Au or Pt have been studied and showed satisfactory selectivity and activity in this reaction [13–15]. The homogeneous catalysts showed higher atomic efficiency but undergo deactivation in aqueous solution, while the heterogeneous

analogues work in any conditions, providing the additional advantage of easy separation, recycling and regeneration if needed.

According to the Sabatier hypothesis and the typical volcano curve of formic acid decomposition, the most ideal catalysts for this reaction are the platinum group metals (PGM), followed by some transition metals in the following order: Pt, Ir, Ru, Pd, Rh, Cu, Ni, Co, Fe and W [16]. The selection of catalysts in the last decade has been conditioned by the price of PGM and the high heat created in the formation of $H_2$ over Ni, Co and W, meaning that it is necessary to find a compromise using one group or another, and the bimetallic formulations represent an interesting alternative. This is why, in view of the PGM price today, the maximization of the noble metal activity per atom is mandatory in order to minimize, as far as possible, the possibility of noble metal loading adding a metal (Fe, Ni or Co) with an active function in the FAD reaction into the catalyst formulation [17–21]. Regardless of the used metal's composition, its distribution on the catalyst' surface increases the availability of the active sites and, consequently, the reaction rate [15]. Supports such as $Al_2O_3$, $CeO_2$, $TiO_2$, graphene and activated carbon [13,18,22,23], usually being carbon-based materials, have been reported as the most suitable supports for the reaction [15,24–27] due to their different hydrophilicities in liquid phase reactions in comparison to the classical mineral oxides.

In addition to the active sites, the FAD reaction in liquid phase is improved by the presence of additives, usually bases or formate salts [20,21,28], acting as reaction intermediates. Their use enables reaction rate control through acid/conjugated base equilibrium, exercising an inhibitory effect of the excess of one type or another on hydrogen production, usually counteracted by the increase in the temperature [29–31]. In addition, hydrogen production is accompanied by $CO_2$, which can remain dissolved as bicarbonate/carbonate species, changing the pH of the final solution [2,32,33]. Hence, the determining reaction parameters are the temperature, pH (formic acid/formate ratio), support and active sites.

In light of the above, this work aimed to study the influence of the active phase's nature on the catalytic behavior of Pd/carbon catalysts. A series of Pd catalysts were prepared and used to study their activity in the FAD reaction as a function of the employed temperature, the presence of a transition metal (cobalt, in our case) and the presence of additives in the reaction medium.

## 2. Results and Discussion

### 2.1. Catalyst Characterization

Table 1 summarizes the ICP-OES measurements of the real metal content. The Pd and Co contents (wt.%) are very similar to the theoretically expected values (values within 10% of the error). The surface area and pore size of the catalysts are also shown in Table 1, suggesting a well-distributed mesoporosity for all the samples, with an average pore size of around 5 nm.

**Table 1.** Catalysts' metal contents and textural properties.

| Catalyst | ICP-OES (wt%) | | Particle Size TEM, nm | | Dispersion Pd | $S_{BET}$ $(m^2/g)$ | $S_{EXT}$ $(m^2/g)$ | Vpore $(cm^3/g)$ | Pore Diameter (Å) |
|---|---|---|---|---|---|---|---|---|---|
| | **Pd** | **Co** | **Pd** | **Co** | | | | | |
| AC | - | - | - | - | - | 844 | 368 | 0.41 | 53.02 |
| Pd 0.5 | 0.5 | - | - | - | - | 852 | 371 | 0.40 | 51.49 |
| Pd 1 | 0.9 | - | - | - | - | 846 | 372 | 0.40 | 51.14 |
| Pd 2 | 1.8 | - | - | - | - | 817 | 384 | 0.45 | 54.52 |
| Pd 5 | 5.1 | - | 1.64 | - | 0.68 | 759 | 354 | 0.41 | 54.61 |
| Pd 10 | 9.6 | - | 5.62 | - | 0.2 | 736 | 316 | 0.36 | 53.82 |
| Pd:Co 5 (3:1) | 4.2 | 0.7 | 1.86 | 18.2 | 0.6 | 808 | 372 | 0.41 | 52.86 |
| Pd:Co 5 (2:1) | 3.7 | 1.2 | 1.94 | 20.8 | 0.58 | 799 | 346 | 0.39 | 52.83 |
| Pd:Co 5 (1:1) | 2.3 | 3.7 | 1.94 | 7.5 | 0.57 | 785 | 349 | 0.40 | 53.46 |
| Pd:Co 5 (1:3) | 1.7 | 3.4 | 1.99 | 30.5 | 0.56 | 787 | 351 | 0.38 | 53.33 |
| Co 5 | - | 5.3 | - | 27.3 | - | 750 | 349 | 0.39 | 53.96 |

All the catalysts presented surface area and pore size values close to those of the parent support, although some deviation could be found after metal deposition. The increase in Pd loading for the monometallic series and the increase in Co loading for the bimetallic series caused the same effect: specific surface area decreases. A significant presence of metal can cause some pore blocking and, as a consequence, diminution in the surface area of the solid.

The X-Ray diffractograms of the monometallic Pd and bimetallic PdCo catalysts are shown in Figures 1 and 2, respectively.

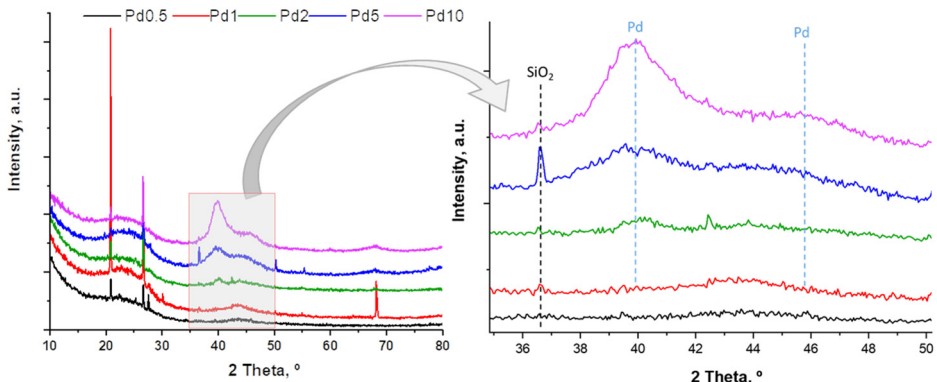

**Figure 1.** XRD patterns of Pd-activated carbon catalyst as a function of palladium loading.

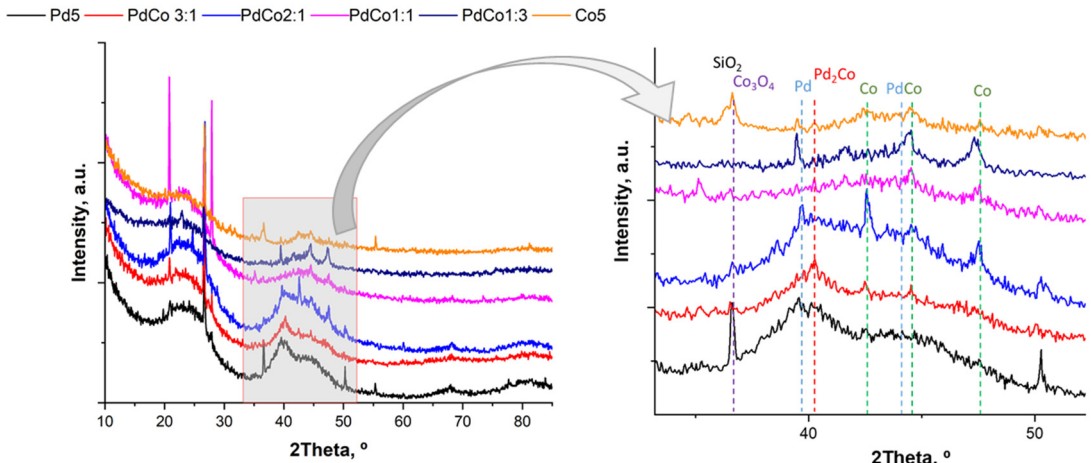

**Figure 2.** XRD patterns of PdCo catalyst as a function of palladium/cobalt ratio.

The signature of the commercial activated carbon is visible in all the samples, presenting two halos at around 23 and 43° 2θ, corresponding to (100) and (002) carbon diffractions, respectively, and associated with the random stacking of graphite sheets. In addition, some sharp diffractions are detected and attributed to the presence of mineral quartz and cristobalite $SiO_2$ phases [34]. This silicon oxide phase is a mineral residue that is present in the commercial activated carbon sample obtained during its production. The presence of metallic Pd is suggested by the diffractions observed at around 40 and 46° 2θ (JCPDS 00-046-1043), which are hardly visible at low loadings but of significant intensity for the Pd 10 sample. Although the intensity of diffractions increases with metal loading, the broad diffractions suggest a low average particle size for all the samples.

The characteristic diffractions of Pd are also visible in the bimetallic samples, especially those with higher Pd/Co ratios (Figure 2). Nevertheless, the slight shift in the main Pd diffractions for the PdCo 3:1 and PdCo 2:1 samples suggests the formation of an intermetallic compound with $Pd_2Co$ stoichiometry accompanied by Pd and/or Co nanoparticles. At low Pd/Co ratios, more signals appear due to Co-containing phases. The monometallic Co

catalyst showed the presence of a metallic Co (JCPDS00-015-0806) and $Co_3O_4$ spinel phase (JCPDS# 00-042-1467).

In general, Pd particle size seems to decrease (suggested by the disappearance of Pd diffraction) with Co fraction increase. Nevertheless, this statement must be confirmed via microscopy, taking into account the fact that at the highest Co loading, the XRD is controlled by the cobalt phases, and in some samples, the formation of $Pd_2Co$ is indicated.

Figure 3 summarizes the TEM analysis performed based on 250 particles in every sample for the bimetallic series and the Pd 5 and Pd 10 catalysts. As we did not detect pure Pd particles with a size superior to 7 nm, we calculated the average Pd particle size in the 0–6 nm range, with the Co/CoOx particles starting from 7 nm or more.

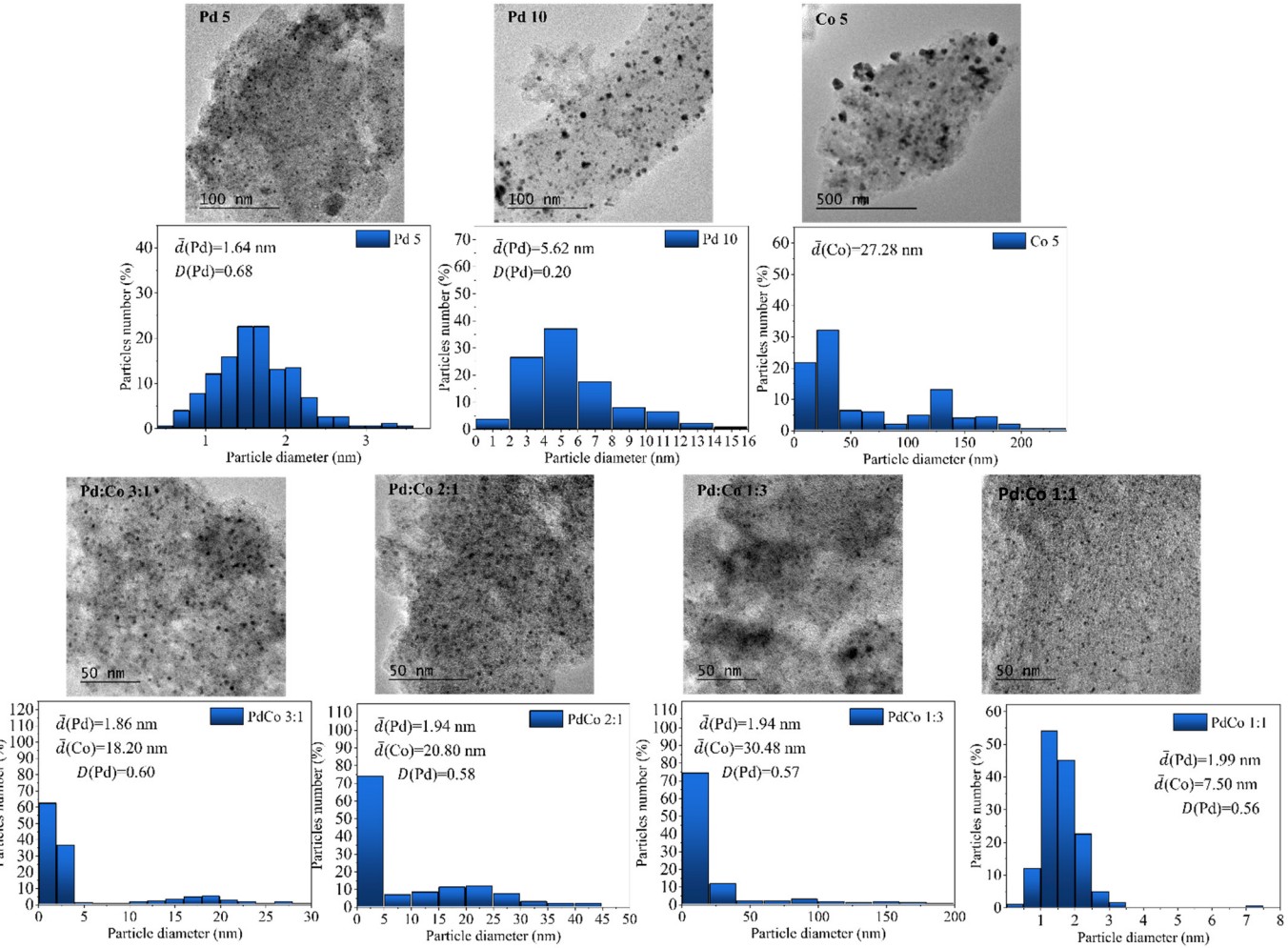

**Figure 3.** HRTEM images and particle size distributions for the studied samples.

A homogeneous monomodal distribution was found for the Pd 5 and Pd 10 samples, with an average particle size of around 1.64 nm for the former and particles almost three times larger (5.62 nm) for the latter. Interestingly, the bimetallic catalysts present a bimodal distribution of small Pd particles and much larger-sized $Co/Co_3O_4$ particles. The Pd average size oscillates around 1.9 nm for all the samples, whereas the Co distribution is relatively heterogeneous, presenting important aggregates [35] over a wide particle range, as shown by the mapping analysis (examples given for the 5Pd and PdCo 1:3 samples in Figure 4). The bimodal distribution allowed us to calculate the particle size of the Pd (all particles within the range of 0–7 nm) and Co (all particles > 7 nm), respectively, under the supposition that all small particles are Pd-rich, as confirmed by the FESEM distribution (Table 1). The Co particles size increases with the loading of PdCo1:1, with

the only exception being cases where the particle size appears homogeneous, without a clear distinction between Pd and Co, suggesting that the homogeneous $Pd_2Co$ phase can be formed in this sample while remaining undetected via XRD. The monometallic Co sample shows a wide heterogeneous distribution with an average particle diameter of 28 nm, very similar to the Co-rich PdCo 1:3 sample. The similar dispersion calculated for the Pd particles in all the 5 wt% containing samples (mono- or bimetallic) suggests that the Pd distribution is less influenced by the presence of a second metal, as in the case of Co.

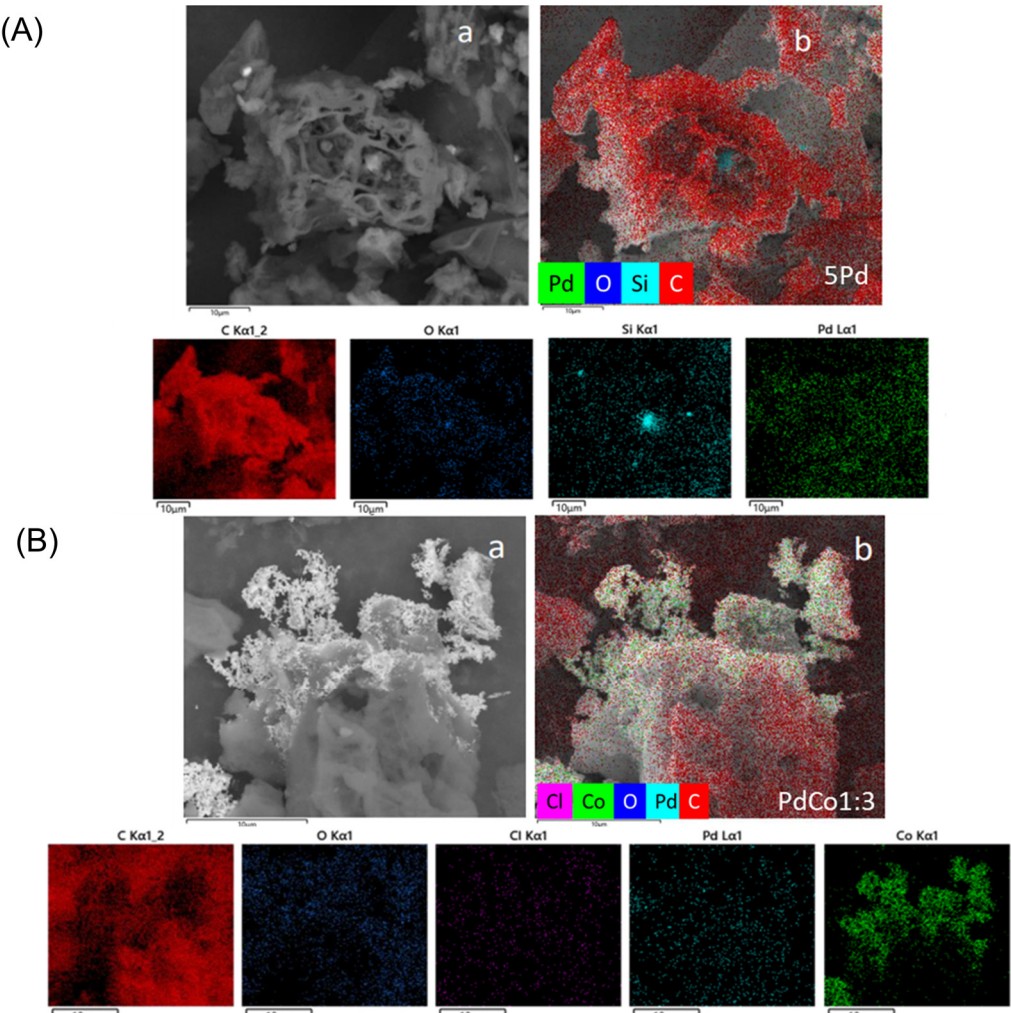

**Figure 4.** FESEM-EDS analysis of (**A**) Pd5 and (**B**) PdCo 1:3 samples. (**a,b**) Electron and EDS layered images, respectively.

## 2.2. Catalytic Activity

Catalyst Screening

The cumulative hydrogen and total gas production for both series (mono- and bimetallic catalysts) are presented in Figure 5. Only $CO_2$ and $H_2$ (Figure 5A,C) were detected during the reaction and represent the total gas production (Figure 5B,D). No CO was observed due to the inhibition of the dehydration reaction in the aqueous phase. The $H_2/CO_2$ ratio remained close to 1 throughout the total time in the stream.

It appears that hydrogen production increases with Pd loading (Figure 5A) to a certain value, with the highest charges being practically equivalent in terms of the total hydrogen production. The initial hydrogen production velocity (measured during the first minutes of the reaction) increases with the Pd charge (Figure 6), with the 10% sample far from showing a linear dependency. The lower initial hydrogen production for the Pd 10 sample, in comparison with the Pd 5 one for the same global production, suggests a different

availability of active sites. Indeed, the highest loading sample presents a larger average particle size, indicating lower Pd surface availability. The presence of linear dependency suggests that the initial velocity of the reaction depends exclusively on the state of the active surface (dispersion, metal distribution), while the total hydrogen production responds to the total Pd loading (Figure 5A,B). The monometallic Co is inactive (not shown) in these particular FAD conditions, but its presence clearly influences Pd behavior in the bimetallic samples. Some metal–metal synergy can be perceived in the PdCo 2:1 and PdCo1:3 samples (the second and third samples in terms of the total hydrogen production in Figure 5B). The formation of the $Pd_2Co$ phase or the configuration of the Pd surface with more $Co/Co_3O_4$ sites appears to influence the initial hydrogen production velocity, the latter being dependent on formic acid adsorption and the products' desorption behavior. It was reported that formic acid interaction with CoOx leads to the formation of metal formate [16], but its decomposition leaves free metal instead of a metal oxide surface. Moreover, the reaction on the Co surface is reported to occur in a mechanism in which the surface formate does not participate in the rate-determining step, in contrast to Pd [16]. One suspects that the presence of metallic Co particles changes the Pd's behavior, leading to a catalyst with high hydrogen production for a low Pd content.

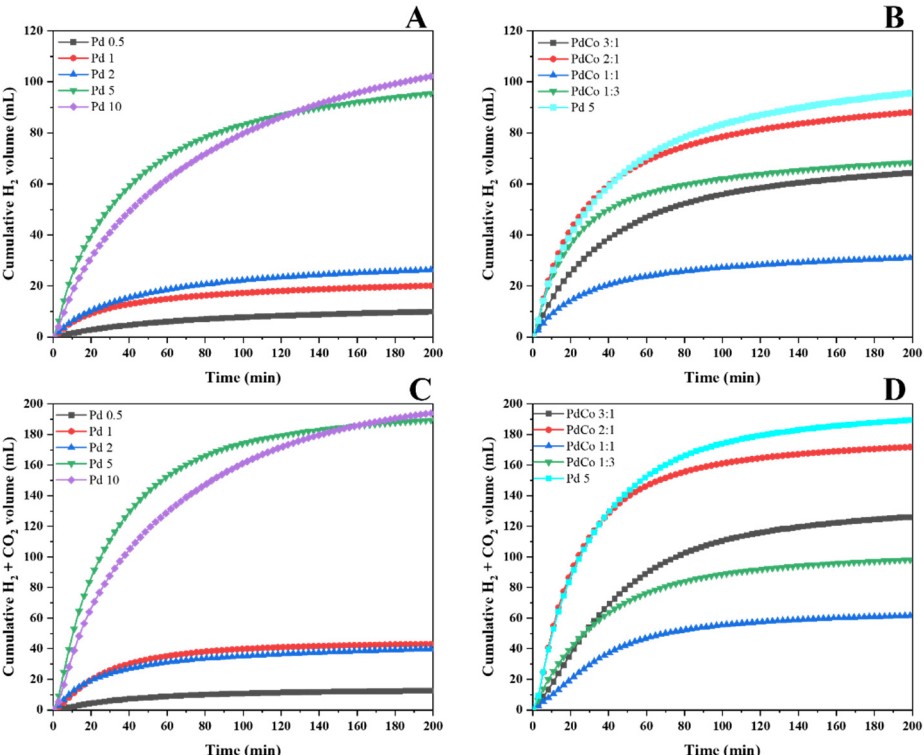

**Figure 5.** Cumulative hydrogen and total gas ($H_2$ + $CO_2$) production as a function of (**A**,**C**) Pd loading and (**B**,**D**) Co presence.

The Pd 5 sample was selected to calculate the apparent activation energy using the measured initial velocity as a function of temperature within the range of 50–65 °C. The activation energy of 60 kJ/mol calculated using the Arrhenius plot coincides with some reports in the literature [36,37]. This value is suggested to vary with Pd particle sizes between 45 and 64 kJ/mol, being greater for larger particles [38].

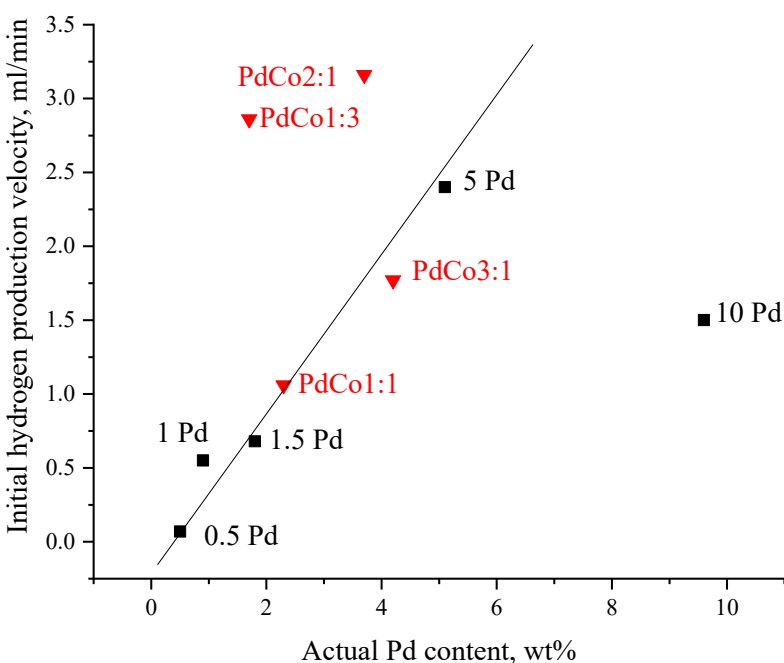

**Figure 6.** Measured initial hydrogen formation velocity as a function of the actual Pd content.

The addition of formates during the FDA reaction is a common way to increase the intermediate concentration on the Pd sites in order to promote hydrogen formation [39,40]. Indeed, the addition of ammonium (AF) and sodium (SF) formates in FA/formate 1:9 molar ratios (FAD reaction with FA + formate 1 M at 60 °C with Pd 5 catalyst) greatly increases hydrogen production (Figure 7). The FA:AF system shows an increment approximately 4 times that of the formic acid 1 M solution and approximately 2.5 times higher than that of the FA:SF system in these conditions. The initial hydrogen formation velocity increases from 2.4 mL/min to 3.5 mL/min and 8.8 mL/min in the presence of SF and AF, respectively. The nature of formate also seems to be important for providing formate ion, through which the reaction proceeds. The rate of the reaction is controlled by the formic acid/formate ratio. At a high ratio, formic acid decomposition is similar to pure acid dehydrogenation, while with the decrease in the ratio, the dehydrogenation reaction is accelerated by the Le Chatellier formic acid/formate equilibrium up to a certain value, after which an inhibitory effect appears with the formation of bidentate formic acid–formate complexes on the catalyst's surface [40]. While for sodium formate, only the increase in the formate concentration affects the catalyst's behavior (Na presence is unimportant) [41], for ammonium formate, the increase in hydrogen production is suggested to occur through a complex mechanism involving amine groups. It has been reported that these groups facilitate formic acid dehydrogenation through O-H bond dissociation and proton elimination, creating the basic environment around the metal catalyst. The latter leads to faster formate CH bond cleavage over the metal catalyst and, as a consequence, higher $H_2$ release [33,42].

As observed in the direct $H_2$ flow curve, the catalyst surface is progressively covered by the reaction intermediates, which leads to a progressive decrease in the gas flow. Nevertheless, the catalyst's activity is completely recovered after the heating of the post-reacted samples at 150 °C in an oven for a few hours. The presence of formates provokes a secondary effect, and the deactivation of the active surface remains incomplete. Neither the presence of sodium nor that of ammonium formates causes the loss of hydrogen production within 200 min, as is the case for the pure formic acid solutions. Some authors have suggested that the formation of hydrogen-bonded complexes of formic acid and formate aid in the stabilization of the latter in monodentate adsorption configurations, facilitating hydrogen production [43,44] and inhibiting bidentate formate adsorption, this being responsible for the deactivation of the active catalytic surface [20,31].

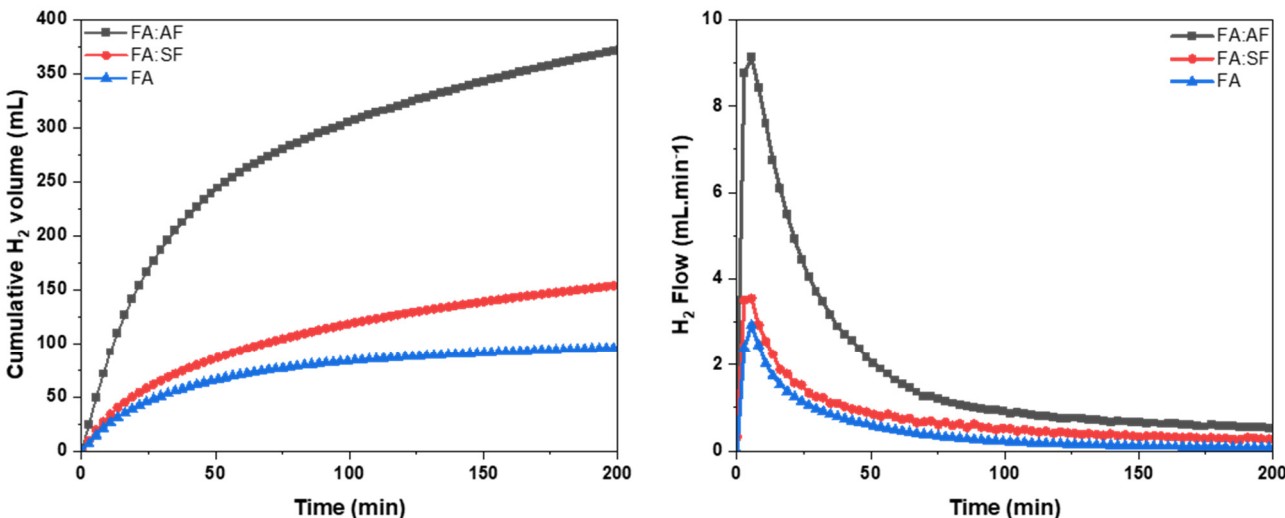

**Figure 7.** Hydrogen production over Pd 5 catalysts in the presence of sodium/ammonium formate.

Table 2 compiles the activity results of different published studies, which were compared to our catalysts in terms of the TON and TOF. Some of the values (TOF or TON) were directly indicated in the studies, while others were recalculated by us using the TOF = TON/t(h) relationship. For the TOF/TON calculation, we considered that the active phase in these conditions is the Pd metal alone, as the monometallic Co sample showed complete inactivity. With this in mind, we used the average particle size of Pd calculated within the 0–7 nm range to calculate metal dispersion and to normalize the activity of the samples over the exposed metal surface.

**Table 2.** Activity results in terms of TON and TOF at 10 min.

| Catalyst | Conditions | Activity | | Ref. |
|---|---|---|---|---|
| | | TON | TOF ($h^{-1}$) | |
| Pd 5 | | 32.3 | 193.7 | |
| Pd 10 | | 35.8 | 214.6 | |
| PdCo 3:1 | | 27.1 | 162.6 | |
| PdCo 2:1 | FA 1 M, 100 mg catalyst, 60 °C | 62.3 | 373.6 | |
| PdCo 1:1 | | 32.1 | 192.6 | This work |
| PdCo 1:3 | | 54.1 | 324.6 | |
| Co 5 | | - | | |
| Pd 5 | FA:AF 1 M (1:9), 100 mg catalyst, 60 °C | 112.5 | 675.0 | |
| Pd 5 | FA:SF 1 M (1:9), 100 mg catalyst, 60 °C | 42.0 | 252.2 | |
| Pd/C 10% | FA 4 M, 100 mg, 60 °C | 56.5 | 339.0 | [37] |
| Pd/mpg-$C_3N_4$ 9.5% | FA 1 M, 50 mg catalyst, 25 °C | 49.8 | 292.8 | [45] |
| Pd/C 9.5% | | 11.9 | 70.0 | |
| Pd/C 5% | FA:SF 2 M (1:9), 20 mg catalyst | 456.6 | 228.3 [a] | [46] |
| $Pd_{0.7}Co_{0.3}$/C 10% | FA 1 M, 25 °C | 38.25 | 15.3 [b] | [19] |
| Pd/C 6.5% | FA 1 M, 50 mg catalyst, 60 °C | 216 | 650 [c] | [38] |

[a] at 120 min, [b] at 150 min, [c] at 20 min.

Taking into account the diversity of catalytic systems that can be found in the literature, our catalytic systems show similar results to those reported for similar systems. It is clear from the table that the best way to increase hydrogen production is to use formic

acid/formate solutions acting directly on the mechanism of reaction and influencing the mode of intermediate adsorption and, as a consequence, the rate of catalyst deactivation. The use of other supports, such as $C_3N_4$, or a second metal also increases the TOF values. The use of Co, in our case, increased the TOF/TON values in two particular metal ratios, with the highest in Co and the PdCo 2:1 relation, for which a mechanism without the participation of formates is suggested. Obviously, the increase in hydrogen production is strictly related to the formic acid/formate adsorption on the Pd surface. On a pure Pd surface, the optimal FA/formate ratios are needed to promote monodentate adsorption in order to decrease the deactivation caused by intermediates, while in PdCo systems, the presence of Co decreases the role of formate as an intermediate. The possible presence of ordered structures, such as $Pd_2Co$ (as suggested by XRD for these two samples), could be responsible for the increase in the initial $H_2$ velocity, indifferent to the presence of sodium formate.

## 3. Materials and Methods

### 3.1. Catalyst Preparation and Chemicals

Commercial activated carbon (AC) DARCO G-60 (Sigma-Aldrich, St. Louis, MO, USA; CAS: 7440-44-0) was used as a support, with Pd (II) (Johnson Matthey, London, UK; purity = 47.15%) and Co (II) acetates (Sigma-Aldrich, St. Louis, MO, USA; purity $\geq$ 95%) as metal precursors.

Following a previous report [24], two series of Pd catalysts, including monometallic (0.5, 1, 2, 5, 10 wt%) and Pd:Co bimetallic catalysts (5 wt% total metal loading with 1:1, 1:3, 2:1 and 3:1 Pd/Co molar ratios), were prepared via wetness impregnation. As a reference, a monometallic 5 wt% Co catalyst was also prepared. The desired quantity of metallic precursor was dissolved in acetone (~20 mL, $1 \times 10^{-3}$ M), and 2 g of support was mixed for 30 min with the precursor solution before solvent evaporation in a rotary evaporator (40 mbar pressure, 40 °C). The final catalysts were dried overnight at 100 °C and reduced prior to use in a $N_2$:$H_2$ (50:50 mL·min$^{-1}$) flow at 350 °C for 2 h at a 5 °C·min$^{-1}$ heating rate. In brief, the support was omitted from the labels, and all the samples were named for the metals to indicate the changing parameters. The monometallic series adopted the nomenclature Pd X, with X being 0.5, 1, 2, 5 or 10, corresponding to the targeted loading of Pd. The bimetallic series was named as PdCo (X:Y), with X:Y corresponding to the molar ratios of the two metals. Thus PdCo (1:3) stands for 5 wt% total metal (Pd + Co) with a Pd:Co molar ratio of 1:3.

### 3.2. Characterization Methods

XRD measurements were performed at room temperature using an X'Pert Pro PANalytical diffractometer with a Cu anode working at 45 kV and 40 mA. The diffractograms were recorded in the continuous scan mode from 35 to 65° 2θ using a 0.03° step size and a step time of 500 s.

The textural properties of the samples were determined via $N_2$ physisorption at 77 K with Micromeritics Tristar II equipment. The samples were previously degassed at 350 °C for 12 h using a Micromeritics 061VacPrep vacuum degasser. The specific surface areas were obtained according to the BET method. The size and pore distributions were calculated based on the desorption isotherms using the BJH method.

The metal contents were measured via Inductively Coupled Plasma Spectroscopy (ICP-OES) using an iCAP 7200 ICP-OES Duo (ThermoFisher Scientific, Waltham, MA, USA) spectrometer. The samples were prepared using an ETHOS EASY microwave digestor (Milestone). For the analysis, 5/10 mg of sample was dissolved in a 3 mL HCl/2 mL $HNO_3$/2 mL $H_2O_2$ mixture under microwave heating at 230 °C for 15 min. Prior to analysis, the resulting solution was diluted with MilliQ water to a final volume of 50 mL.

The metal particle size distributions were determined using High-Resolution Transmission Electron Microscopy (HR-TEM) measurements with a JEOL 2100Plus (200 kV) microscope equipped with an LaB6 filament. Digital images were taken with a CCD

(Gatan) camera. The samples were deposited over copper grids. The mean particle size was estimated according to Equation (3) based on 200 single-particle measurements from TEM images.

$$D_p = \frac{\sum d_i \cdot n_i}{n_T} \tag{3}$$

where $D_p$ is the average particle size, $d_i$ is the geometric diameter of the $n_i$ particles with the same diameter, and $n_T$ is the total number of particles.

### 3.3. Catalytic Tests

The FAD reaction was carried out in a round, four-neck, glass semi-batch reactor (250 mL) equipped with a gas inlet/outlet (100 mL·min$^{-1}$ N$_2$ flow, used as carrier), previously described by Santos et al. [24]. In a typical experiment, the reactor was charged with 100 mL formic acid solution (1 M) and maintained under vigorous stirring while the flowing nitrogen removed any remaining oxygen from the system. Once the reaction temperature reached the desired value (typically 60 °C), 0.1 g of catalyst was added, considering this moment as the initial time of reaction (total time in the stream: 200 min). The outlet gas was analyzed continuously using a Varian CP-4900 gas chromatograph equipped with a TCD detector and molecular sieve 5 A column, using N$_2$ as the internal standard for quantification purposes. CO$_2$ was analyzed separately with a CO$_2$ analyzer, Vaisala MI70, coupled to the micro-GC.

The catalytic activity was compared in terms of the turnover number (TON) and turnover frequency (TOF), calculated on the basis of the surface-available Pd according to Equations (4) and (5), respectively:

$$TON = \frac{mmol\ H_2\ produced}{mmol\ Pd \times D} \tag{4}$$

$$TOF = \frac{TON}{t(h)} \tag{5}$$

where ($t$) is the reaction time, the *mmol Pd* was determined via ICP-OES, and the dispersion ($D$) was calculated according to Equation (6) [47] using the volume occupied by an atom of Pd, denoted as $v_m$ of the area $a_m$, with $D_p$ as the average size. The reaction time was considered only in the TOF parameter calculation.

$$D = \frac{6 \cdot \left( \frac{v_m}{a_m} \right)}{D_p} \tag{6}$$

The TON and TOF expressions were used to compare the catalytic activity with that of the similar catalysts reported in the literature.

## 4. Conclusions

It is confirmed that favorable results were obtained for hydrogen storage in the formic acid dehydrogenation reaction using Pd-based catalysts. These systems can be modified in order to increase hydrogen production to the greatest extent possible for the lowest possible Pd concentration. The bimetallic approach presents an interesting alternative for noble metal loading decrease. Nevertheless, only some specific metal combinations result in high metal synergy and an influence on hydrogen productivity. The presence of a significant concentration of Co/CoOx or the formation of an ordered Pd$_2$Co structure diminishes the importance of formate intermediates in the reaction, leading to a great increase in hydrogen production per mol of Pd.

Another way to increase the hydrogen concentration is to use formic acid/formate solutions in order to stabilize the monodentate adsorption of complexes formed by these molecules. In this way, the deactivation rate is decreased, leading to stable hydrogen production over a long period of time.

**Author Contributions:** Conceptualization, S.I.; methodology, E.R.-L., M.I.D. and S.I.; validation, S.I. and M.A.C.; formal analysis, M.I.D.; investigation, M.R.P., E.R.-L.; resources, S.I. and M.A.C.; data curation, M.I.D., S.I. and M.A.C.; writing—original draft preparation, M.R.P.; writing—review and editing, M.R.P., E.R.-L., M.I.D., S.I. and M.A.C.; supervision, M.A.C.; project administration, M.A.C.; funding acquisition, S.I. and M.A.C. All authors have read and agreed to the published version of the manuscript.

**Funding:** Financial support was obtained from the Spanish Ministerio de Ciencia e Innovación (MCIN/AEI/10.13039/501100011033/) and the FEDER Funds una manera de hacer Europa (Projects ENE2017-82451-C3-3-R and PID2020-113809RB-C32). In addition, the financial support received from Junta de Andalucía via Consejería de Transformación Económica, Industria, Conocimiento y Universidades and its PAIDI 2020 program (grant P18-RT-3405), co-financed by FEDER funds from the European Union, is highly appreciated.

**Data Availability Statement:** Available upon request.

**Acknowledgments:** E. Ruiz-López would like to acknowledge the Spanish Ministerio de Universidades and the Unión Europea—NextGenerationEU for the financial support of her Margarita Salas Fellowship.

**Conflicts of Interest:** The authors declare no conflict of interest.

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
