# Peer review of "Formic Acid Dehydrogenation over a Monometallic Pd and Bimetallic Pd:Co Catalyst Supported on Activated Carbon"

_catalysts, doi:10.3390/catal13060977_

Round 1
Reviewer 1 Report
General Comments: In this work, Co introduction allows palladium to lower use for more efficient formic acid dehydrogenation, due to the formation of Pd2Co phase or the decoration of the Pd surface with more Co/Co3O4 sites and extra two kinds of formate in the FA system. Some key points and experimental description are not accurate and scientific. Therefore, there are still some issues to be resolved before its publication.
Comment 1. The adjective “difficult” is used incorrectly in the first paragraph of the Introduction. Please check and correct it.
Comment 2. The magnification of 35-50°XRD patterns in Figure1 and 35-55°XRD patterns in Figure 2 are not easily seen. It is better to show two parts (for example, 10-80°and 35-50°in Figure 1) with two clear pictures respectively.
Comment 3. Due to the lack of appropriate characterization method, the distribution of Co and Pd on these catalysts is not clear, which influences the analysis of particle sizes, particle composition and catalytic activity differences. You are supposed to take EDS mapping test to analyze the Co and Pd elements’ distribution.
Comment 4. The author writes that the increase of the formate ions concentration affects the catalyst behavior, while the existence of Na+ seems unimportant. However, it can’t be proved from Figure.6 and other Figures or Tables, could the author supplement the test results to prove this conclusion?
Comment 5. The particles size distribution of PdCo 1:1 in Figure 3 does not show the particle diameter distribution more than 4nm. However, the author notes that the average diameter of Co is more than 7nm, dose it indicate that the particle diameter distribution is not matched? If so, please refine this figure or explain why this figure is unmatched.

Minor editing of English language required
Reviewer 2 Report
This research investigated formic acid dehydrogenation over Pd or Pd-Co supported on carbon catalysts. The authors described Co can improve catalytic performance in formic acid dehydrogenation, but there are not much proper data and mechanism. So, this article can publish after revising all comments.
Introduction
- The author mentioned homogeneous and heterogeneous catalysts in formic acid dehydrogenation. What is the difference (advantage or disadvantage) in formic acid dehydrogenation? Why are you want to use heterogenous catalysts in formic acid dehydrogenation? You should describe the proper information.
- 16-20, references/ Author mentioned Fe, Ni, Co transition metals effect. However, it is hard to understand what the point is. You should describe detailed information. (e.g, electronic effect or synergetic effect)
- 15, 23-26 reference/ You should describe detailed information about carbon supports (hydroxylation).
- You must mention what is the new finding of this research. Also, you need to write this part in abstract.
Experimental
- Preparation of catalyst/What are the evaporation temp and pressure? Are you using a rotary evaporator?
- Metal dispersion equation; you need to describe in detail information what is the vm , am , Dp. I know that there are specific numbers, so you should mention this number.
- You should write more detail information. How to prepare the ICP samples(e.g microwave, solvent, etc).
Results and discussion
- You should have raw Carbon XRD data, Why does the catalyst have SiO2 phase?
- The intensity of Pd peaks is different. I recommend calculation of crystallite size. What is the proper reason for low average particle size? / Figure is not well organized. You should insert all Pd peaks in XRD graph by using the mark (#, *, etc)
- You calculated metal dispersion for TON. You should insert metal dispersion in Table 1 and properly describe with TEM images.
- How can you estimate TON or TOF? You can estimate only pure Pd/C catalyst. In Pd-Co/C catalyst may have Pd, Pd-Co, and Co particles. In this case, how can you measure the metal dispersion?
- Dispersion from TEM is not correct. You may use CO-Chemisorption.
- TEM; What is the proper reason about Pd particle size seems to decrease?
- This research must have XPS data. You mentioned Pd-Co structure is important for formic acid dehydrogenation, but how can you explain only XRD data?
- Figure 4, You mentioned, “The lower initial hydrogen production for the Pd 10 sample in comparison with the Pd 5 one for the same global production suggests a different availability of the active sites.” What are the different active sites? Can you explain?
- You mentioned “The formation of Pd2Co phase or the decoration of the Pd surface with more Co/Co3O4 sites appears to influence the initial hydrogen production velocity, being the latter dependent on the for[1]mic acid adsorption and products desorption behavior”, How is the different? What is the mechanism? Can you describe the proper mechanism?
- Figure 6, You should mention the proper mechanism in FA, FA-SF, and FA-AF dehydrogenation.
- Formic acid dehydrogenation can produce CO2 and CO as a by-product. However, there is no information about CO2, CO, and H2 concentration (or purity).
- Deactivation (Figure 6). I recommend a regeneration test. There are some regeneration methods. Before or after the regeneration, You should have XRD (for crystallite size) and TEM(for dispersion) data.
N/A
Round 2
Reviewer 2 Report
Results and discussion
1. You should have raw Carbon XRD data, why does the catalyst have SiO2 phase? The activated carbon is a commercial sample prepared from a residual biomass and always contains mineral component (SiO2 issue from the natural abounding silicates in the biomass). It was not included in the XRD data for a sake of simplicity. Anyway we have published in other studies the XRD data (DOI: 10.1016/j.cattod.2023.01.019; DOI:10.1016/j.apcatb.2021.119938 or reference 35 in the present study) of the bare commercial Darco sample, where you can find that the sample contains always that SiO2 fraction.
è You should describe the SiO2 phase from Carbon in manuscript.
4. How can you estimate TON or TOF? You can estimate only pure Pd/C catalyst. In PdCo/C catalyst may have Pd, Pd-Co, and Co particles. In this case, how can you measure the metal dispersion? As explained above and as confirmed by the FESEM mapping distribution the Pd and Co sizes are calculated considering that the first part of the bimodal distribution pertains to Pd and the second one to Co. It is not so erroneous to consider taking into account the FESEM mapping analysis. Considering this, the dispersion was calculated and also TOF and TON. & 5. Dispersion from TEM is not correct. You may use CO-Chemisorption. You are right that the total dispersion must be calculated on the totality of particles presented on the surface and on the basis of CO chemisorption. However, the absence of activity for the pure Co catalysts allowed us to consider that the active metal is Pd which is rather well dispersed and we have use its average size for the calculation of the dispersion.
è You should describe this information in manuscript (how to measure metal dispersion, what was the considering factor for calculation of TON and TOF).
10. Formic acid dehydrogenation can produce CO2 and CO as a by-product. However, there is no information about CO2, CO, and H2 concentration (or purity). & 11. Deactivation (Figure 6). I recommend a regeneration test. There are some regeneration methods. Before or after the regeneration, You should have XRD (for crystallite size) and TEM(for dispersion) data.
I strongly recommend inserting the figure (related to question number 10 and deactivation results) in the manuscript. Authors describe our (Pd-Co) catalysts are good in formic acid dehydrogenation. However, what is the good point for other researchers? The finding is important, but it is more important to raise the problem or limitation of your catalyst by providing deactivation data.
